# Learning efficient hybrid particle-continuum representations of non-equilibrium N-body systems

## Abstract

An important class of multi-scale, non-equilibrium, N-body physical systems deals with an interplay between particle and continuum phenomena. These include hypersonic flow and plasma dynamics, materials science, and astrophysics. Hybrid solvers that combine particle and continuum representations could provide an efficient framework to model these systems. However, the coupling between these two representations has been a key challenge, which is often limited to inaccurate or incomplete prescriptions. In this work, we introduce a method for Learning Hybrid Particle-Continuum (LHPC) models from the data of first-principles particle simulations. LHPC analyzes the local velocity-space particle distribution function and separates it into near-equilibrium (thermal) and far-from-equilibrium (non-thermal) components. The most computationally-intensive particle solver is used to advance the non-thermal particles, whereas a neural network solver is used to efficiently advance the thermal component using a continuum representation. Most importantly, an additional neural network learns the particle-continuum coupling: the dynamical exchange of mass, momentum, and energy between the particle and continuum representations. Training of the different neural network components is done in an integrated manner to ensure global consistency and stability of the LHPC model. We demonstrate our method in an intense laser-plasma interaction problem involving highly nonlinear, far-from-equilibrium dynamics associated with the coupling between electromagnetic fields and multiple particle species. More efficient modeling of these interactions is critical for the design and optimization of compact accelerators for material science and medical applications. Our method achieves an important balance between accuracy and speed: LHPC is $8\times$ faster than a classical particle solver and achieves up to 6.8-fold reduction of long-term prediction error for key quantities of interest compared to deep-learning baselines using uniform representations.

## 1 Introduction

The dynamics of physical systems is often nonlinear and involves the competition of different processes across a wide range of spatial and temporal scales. This gives rise to local non-equilibrium conditions (in the thermodynamic sense) that results in the failure of common numerical approaches. While continuum models (*e.g.*, based on fluid equations) can provide accurate and computationally efficient descriptions of near-equilibrium systems at large scales, they break down when significant departures from local equilibrium are encountered (often at small scales) and give rise to important N-body phenomena. Kinetic (*e.g.*, particle-based) numerical methods can accurately describe these non-equilibrium phenomena but are very computationally intensive, limiting their practical application to small scales. Over the last decades, this has motivated efforts to develop hybrid algorithms that can more efficiently couple continuum and particle representations in a variety of fields, including hypersonic gas dynamics (Schwartzentruber & Boyd, 2006), high-energy-density physics (Fiuza et al., 2011), and plasma physics (Bai et al., 2015).

Plasmas — hot ionized gases of charged particles — are a particularly challenging class of complex physics systems, where long-range electromagnetic interactions inevitably drive multi-scale and far-from-equilibrium dynamics. Indeed, plasma research associated with controlled nuclear fu-

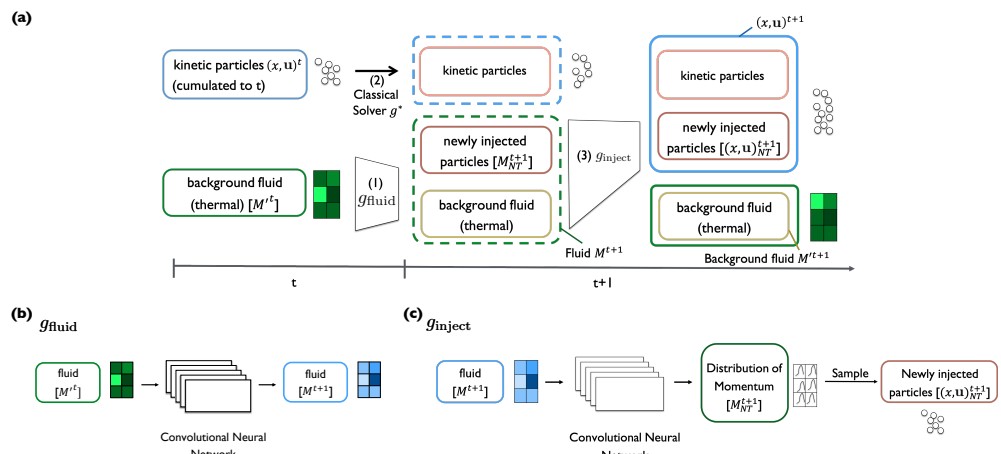

Figure 1: (a) Schematic of our LHPC architecture. It consists of three components: (1) a neural network $g_{\text{fluid}}$ for evolution of thermal sub-population with fluid representation; (2) a solver $g^*$ for the evolution of the non-thermal energetic sub-population with particle representation; (3) neural networks that model the injection of particles from fluid state ($g_{\text{inject}}$). (b) and (c) illustrate the architectures of the components (1) and (3).

sion (Zylstra et al., 2022), advanced laser-plasma particle accelerators (Haberberger et al., 2012), and most space and high-energy astrophysical environments (Drake et al., 2006) have long stimulated the development of hybrid particle-continuum representations to accurately and efficiently capture the nonlinear dynamics of these systems (Winske, 1985; Fiuza et al., 2011; Bai et al., 2015; Liu et al., 2022). In all these cases, small-scale kinetic processes can accelerate a very small group of particles to energies significantly above the mean (thermal) energy, driving the system out of equilibrium. Importantly, this small group (few %) of non-thermal particles can carry away a large fraction (up to 50%) of the system energy and thus impact its global evolution. This motivated the recent development of hybrid representations that use a fluid solver to model the near-equilibrium (thermal) part of the particle distribution and a particle-based kinetic solver to model the evolution of high-energy particles (Kowal et al., 2011; Bai et al., 2015; Guidoni et al., 2016). However, the coupling of the two representations has been based on over-simplified phenomenological prescriptions that can limit their validity and applicability.

In this work, we introduce a method for Learning Hybrid Particle-Continuum (LHPC) models to address the key challenge of efficiently and accurately coupling fluid and kinetic representations of far-from-equilibrium N-body systems. LHPC combines a classical particle-in-cell (PIC) solver to advance the non-thermal particle distribution with a neural network surrogate model that efficiently advances the thermal component using a continuum representation. Most importantly, our key contribution is the use of an additional neural network to learn the self-consistent particle-continuum coupling: the dynamical exchange of mass, momentum, and energy between the particle and continuum representations. This coupling is learned from the data of first-principles PIC simulations, providing an accurate and physics-informed description that addresses the main limitation of previous hybrid methods.

While the combination of classical numerical solvers and deep learning has been explored before (Um et al., 2020; Vlachas et al., 2022), these have been primarily based on uniform representations. The use of machine learning techniques to learn efficient and accurate coupling between continuum and particle representations introduced in this work is a promising and important route for addressing the multi-scale and multi-physics challenge of modeling N-body non-equilibrium systems.

We demonstrate our method on a challenging non-linear, far-from-equilibrium N-body system: the interaction of an intense laser with a solid-density plasma and resulting particle acceleration. Present-day high-power lasers reach intensities in excess of $10^{20}\text{W/cm}^2$, which nearly instantaneously vaporize and ionize solid-state matter upon interaction, resulting in high-energy-density plasma. These interactions give rise to nonlinear and kinetic processes that establish strong elec-

tric fields in the plasma and accelerate particles to high energies over very compact distances. In the last two decades, there has been great interest in exploiting these laser-plasma based accelerators for a wide range of applications, including material science (Patel et al., 2003), imaging (Rygg et al., 2008), and medical therapy (Kroll et al., 2022). Due to the need to capture detailed kinetic physics associated with these interactions, numerical modeling has relied primarily on fully kinetic PIC simulations, which are too computationally intensive – in fact, one-to-one modeling of experimental systems is computationally prohibitive even on the largest supercomputers. Effective hybrid particle-continuum representations have not been previously established for this problem. This is thus a prime example for testing our model and evaluating the ability of LHPC to learn efficient and accurate hybrid representations that can have a transformative impact in the modeling of these complex laser-plasma interactions.

Our results show that LHPC achieves an important balance between accuracy and speed: LHPC is 8 times faster than a classical particle solver, and achieves up to 6.8-fold reduction of long-term prediction error for key quantities of interest when compared to deep-learning baselines using uniform representations. This opens up a novel route to develop more efficient and accurate particle-continuum descriptions across different domains involving non-equilibrium N-body systems, and represents a key step for data-driven multi-scale algorithms for nonlinear physical systems.

## 2    RELATED WORKS

Deep learning has recently emerged as a powerful tool to complement Um et al. (2020) or serve as a surrogate for classical solvers (Sanchez-Gonzalez et al., 2020). They accelerate simulation of physical systems through coarser spatial resolution (Um et al., 2020; Kochkov et al., 2021) or temporal intervals (Li et al., 2021; Vlachas et al., 2022), explicit forward method (Tang et al., 2020; Wu et al., 2022b), or via reduced representations (Sanchez-Gonzalez et al., 2020; Wu et al., 2022a). Two important examples are LED (Vlachas et al., 2022) and solver-in-the-loop (Um et al., 2020). The former optimizes the temporal evolution by employing a solver to simulate the full system in some interval of time and uses latent evolution to evolve in other parts of time, with a pre-defined alternative schedule. The latter uses a fluid solver on a coarser spatial grid and a neural network to correct for the fluid solver's error. However, almost all works employ a uniform representation, based on either all particles or all fluid. This is not appropriate when trying to optimize far-from-equilibrium, N-body systems where a particle description is too computationally expensive for the near-thermal population (where most particles reside) and a fluid description cannot accurately describe the complex distributions associated with the far-from-equilibrium population. Our method aims to address this long-standing challenge by learning an accurate and efficient coupling between different representations involving different physical processes and scales, namely a hybrid particle-continuum representation which is critical to describe non-equilibrium N-body systems. To the best of our knowledge, machine learning methods have not yet been used to address this problem.

## 3    PROBLEM SETUP AND PRELIMINARIES

**Task setup**. In general, simulations of the time-evolution of a dynamical system can be described as follows. The state of the system at time $t$ is $s^t$, and there exists a ground-truth evolution $g^*$ that evolves this state:

$$s^{t+1} = g^*(s^t), \, t = 0, 1, 2, \ldots \tag{1}$$

The ground-truth evolution $g^*$ can either be the physical world which is challenging to predict, or a first-principles numerical solver which can be slow and expensive for large-scale systems. Assume that we have a pre-defined mapping $h$ which maps the state $s^t$ to a representation $S^t$: $S^t = h(s^t), \, t = 0, 1, 2, \ldots$, such that $S^t$ captures the essence of the system. For example, consider an original state $s^t$ represented by particles, and $S^t$ a fluid representation which describes the system with statistics of the particles within each cell (see below). Alternatively, $h$ may be an identity mapping resulting in $S^t = s^t$. A hybrid $h$ mapping will apply both of these representations to different groups of particles. Given the states $\{s^t\}, t = 1, 2, \ldots, T$, the task is to design a proper representation $S^t = h(s^t)$, and learn a surrogate model $g_\theta$, typically in terms of a neural network, which approximates the ground truth evolution of $S^t$:

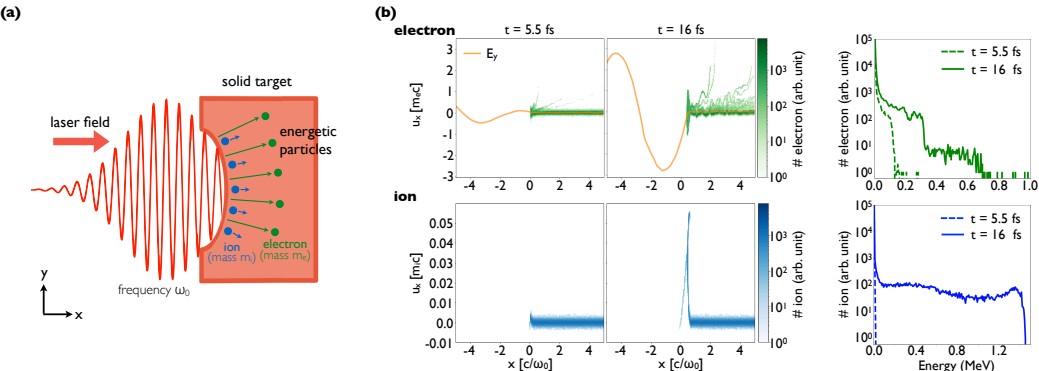

Figure 2: (a) Intense laser-plasma interactions generate energetic particles with rich dynamics. (b) Projection of the particle dynamics onto the longitudinal momentum ($u_x$)-position ($x$) phase spaces, the laser E-field (orange), and the corresponding energy spectra of the particles at two different times.

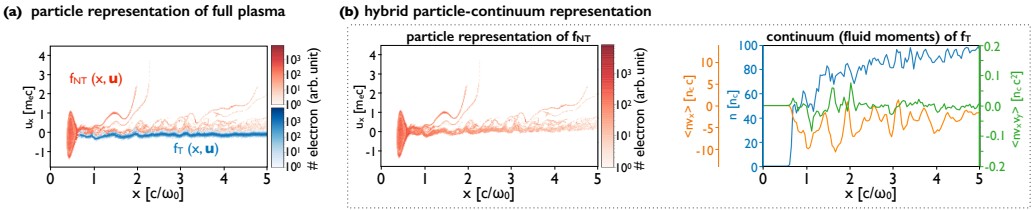

Figure 3: Representations of plasma illustrated with electrons. (a) Particle representation of the non-thermal ($f_{NT}$) and thermal ($f_T$) populations. (b) Hybrid particle-continuum representation showing the same particle representation of $f_{NT}$ and example velocity moments of the thermal electron fluid.

$$\hat{S}^{t+1} = g_\theta(S^t),\ t = 0, 1, 2, \ldots \tag{2}$$

Here $\hat{S}^{t+1}$ is the model $g_\theta$'s prediction of $S^{t+1}$. Our goal is to have the long-term autoregressive evolution of the learned surrogate model $\hat{S}^{t+k} = g_\theta \circ \ldots \circ g_\theta(S^t)$ (composing $k$ times) to closely match the ground-truth $S^{t+k}$.

**Physical problem overview**. In this work, we consider the interactions of intense laser pulses with plasmas, where the laser field accelerates the plasma electrons near the front surface, which in turn sets up a charge separation, generating electric (E-) and magnetic (B-) fields that accelerate and deflect the ions (Fig. 2; see Appendix A for details). These interactions of the charged particles with E- and B-fields (collectively EM-field) occur on ultra-fast time scales ($\sim 10^{-15}$ s), and can produce electrons and ions with energies exceeding $10^6$ electronvolt ($\equiv 1\,\mathrm{MeV}$) – with the electrons moving at close to the speed of light $c$. These are prime examples of non-equilibrium systems, for which computationally expensive first-principles simulations are necessary to accurately capture the associated non-linear dynamics.

**Particle and continuum representations**. The two dominant representations of the plasma system considered in our work are particle (kinetic) and continuum (fluid) representations. A particle representation $\{(x_i^t, \mathbf{u}_i^t, Q_i)\}$ describes the position $x_i^t$, momentum vector $\mathbf{u}_i^t$, and other static properties $Q_i$ (e.g., mass or charge) of each particle $i$. The particles interact with each other, and change their position and momentum accordingly. In contrast, a fluid representation $M^t$ is defined on a fixed grid or mesh. The values on each vertex represent the moments (statistical averages) of the momenta of the particles residing around the vertex. These statistical quantities summarize the states of a large number of particles, permitting a more compact representation. Their physical meaning is well established. For example, the first three moments correspond to the fluid observables mass density, fluid velocity, and pressure. We note that in general an exact representation of the full par-

ticle distribution requires an infinite number of fluid moments (see Appendix A.1). However, if the particle distribution is near local thermodynamic equilibrium it can be described by a compact set of fluid moments. A hybrid particle-continuum model aims to provide an efficient description of a more general distribution of particles by coupling a particle representation for far-from-equilibrium particles with a continuum representation for particles near equilibrium.

**Thermal and non-thermal population**. In this context, we refer to the group of particles in a near equilibrium state as *thermal*, for which the momentum distribution can be well described by a Gaussian (*i.e.* requiring only the mean and variance). On the other hand, we refer to the particles that are far from equilibrium as *non-thermal*. These are the outliers from this Gaussian distribution. In terms of the total particle distribution function, $f$, this allows us to decompose it into a *thermal*, $f_{\mathrm{T}}$, and *non-thermal*, $f_{\mathrm{NT}}$ components, with $f = f_{\mathrm{T}} + f_{\mathrm{NT}}$. See Fig. 3 for an illustration of this separation in physical problem of intense laser-plasma interactions considered in this work; details of the separation algorithm are presented in Appendix B.

## 4 METHOD

### 4.1 METHOD OVERVIEW

Here we introduce Learning Hybrid Particle-Continuum Models (LHPC) to address the key challenges in hybrid representations of non-equilibrium dynamical systems. Specifically LHPC learns two key components, for the evolution of the fluid, and coupling between the particle and fluid representations. In Figs. 1, 3 and the following sections, we detail the data generation and preparation procedure, the main architecture, and its application to the study of intense laser-plasma interactions.

#### 4.1.1 DATA GENERATION AND PREPARATION FROM FIRST-PRINCIPLES SIMULATIONS

Although computationally expensive, full PIC simulations provide ground truth data from first-principles particle-based evolution of plasma dynamics. We run full PIC simulations of laser-plasma interactions in a one spatial dimension, three velocity dimension (1D3V) setting, where the laser intensity is varied to generate multiple trajectories. A trajectory is composed of a simulation with 2000 time steps of evolution, involving 1.5 million (numerical) particles evolving on a $\simeq$10k-cell grid, and is 200 GB in size (see Appendix A.3 for details).

To train the two learned components of LHPC, we transform the ground truth particle data into the fluid and particle representations at each time step. This is achieved by first applying an algorithm that separates the particles into thermal and non-thermal populations (*e.g.* Fig. 3; for details see Appendix B). The labeled particle data is then transformed, with the non-thermal population keeping the original particle representation, and using the velocity moments (up to 2nd order) to represent the thermal fluid. As the system evolves, a small fraction of the particles from the thermal population will gain a significant amount of energy to get detached from the thermal distribution, becoming non-thermal. They are said to be "injected" into the non-thermal population. The transformed ground truth data therefore provides the information of the evolution of both representations, and the coupling (injection) between them.

#### 4.1.2 ARCHITECTURE

LHPC evolves the system state $S^t$ at time $t$ to state $S^{t+1}$ at time $t+1$ with three components (Fig. 1):

(1) A neural network $g_{\mathrm{fluid},\theta}$ with parameters $\theta$ which evolves the (thermal) fluid (Fig. 1b).

(2) A ground-truth solver $g^*$ which evolves the non-thermal particles.

(3) A neural network $g_{\mathrm{inject},\varphi}$ with parameters $\varphi$ which models the interplay of the fluid and particle populations (Fig. 1c). In particular, it models the injection of new non-thermal particles from the fluid, and updates the existing fluid and particle populations accordingly.

At inference, these processes are iterated autoregressively to predict the state of the system into the long-term future.

## 4.2 LHPC FOR LASER-PLASMA INTERACTIONS

Here, we employ LHPC to model laser-plasma simulations. The space in $x$ is divided into $C$ consecutive cells: $c \in [c_{\text{left}}, c_{\text{left}+1}, \ldots, c_{\text{right}}]$ in the $x$ direction. The state $S^t$ consists of:

$$S^t := ((M^t, M'^t, M_{\text{NT}}{}^t), (x_{\text{NT}}{}^t, \mathbf{u}_{\text{NT}}{}^t, x_{\text{PIC}}{}^t, \mathbf{u}_{\text{PIC}}{}^t), (E^t, B^t))_c :$$
$$c \in [c_l, c_{l+1}, \ldots, c_r]$$

where the velocity moments of different populations $M$ and the non-thermal particles $(x, \mathbf{u})$ interact through the fields $(E, B)$. The full pipeline in Fig. 8 in Appendix C further visualizes the dependencies amongst these components and documents what each quantity represents. The following sequence of operations is used to evolve the system from time $t$ to time $t + 1$:

$$M^{t+1} \leftarrow g_{M,\theta}((E, B)^t, M'^t)$$
  Advance fluid moments according to electromagnetic (EM-) field on the grid.    (1a)
  $g_{M,\theta}$ is an instantiation of the $g_{\text{fluid},\theta}$ introduced in Sec. 4.1.2.

$$M_{\text{NT}}^{t+1} \leftarrow g_{M_{\text{NT}},\varphi}(M^{t+1}, M'^t)$$
  Compute moments of new non-thermal particles to be injected    (1b)
  $g_{M_{\text{NT}},\varphi}$ is an instantiation of the $g_{\text{inject},\varphi}$ introduced in Sec. 4.1.2.

$$(x, \mathbf{u})_{\text{NT}}^{t+1} \leftarrow \mathcal{N}(M_{\text{NT}}^{t+1})$$
  Inject new non-thermal particles by sampling from distribution with $M_{\text{NT}}^{t+1}$    (1c)

$$M'^{t+1} \leftarrow (x_{\text{NT}}, \mathbf{u}_{\text{NT}}, M)^{t+1}$$
  Update fluid moments to conserve mass, momentum, and energy    (1d)

$$(x, \mathbf{u})_{\text{PIC}}^{t+1} \leftarrow \left[ G((x, \mathbf{u})_{\text{PIC}}^t, (E, B)^t); \quad (x, \mathbf{u})_{\text{NT}}^{t+1} \right]$$
  Advance existing particles and append newly injected non-thermal particles    (1e)

$$J^{t+1} \leftarrow (M', x, \mathbf{u})^{t+1}$$
  Deposition of electric current $J$ from particles and fluid to grid    (1f)

$$(E, B)^{t+1} \leftarrow J^{t+1}$$
  Advance EM-field on the grid using Maxwell's equations    (1g)

The operators $g_{M,\theta}$ and $g_{M_{\text{NT}},\varphi}$ are instantiated from $g_{\text{fluid},\theta}$ and $g_{\text{inject},\varphi}$ and are deep neural network models learned from data. The operator $G$ and the field solver advances the non-thermal particles with the classical PIC solver. A more detailed description of each component $a$–$g$ can be found in Appendix C.1, and detailed architecture, learning procedure and objective is in Appendix E.

### 4.2.1 BASELINES

We compare with state-of-the-art deep learning-based surrogate model of Fourier Neural Operator (FNO; Li et al. 2021), and a baseline CNN that has the same architecture as our fluid evolution model without hybrid representation. Both architectures are evaluated on two representations: (1) an all fluid representation which models all particles as fluid defined on each cell; (2) a bi-Gaussian fluid representation, that models thermal particles and non-thermal particles separately but each as a Gaussian. Baseline (1) treats the entire system as a fluid and advances the system by learning $g$ as:

$$M^{t+1}, (E, B)^{t+1} = g((E, B)^t, M^t).$$

The improved "bi-Gaussian" baseline (2) models the thermal population and the non-thermal particles each as a Gaussian, and advances the system by learning $g$ as:

$$M^{t+1}, M_{\text{PIC}}^{t+1}, (E, B)^{t+1} = g((E, B)^t, M^t, M_{\text{PIC}}^t).$$

To evaluate the importance of the coupling network $g_{M_{\mathrm{NT}},\varphi}$, we also compare our hybrid model with an ablation where we remove $g_{M_{\mathrm{NT}},\varphi}$, while keeping all other parts the same ("LHPC (no-coupling)"). For all models, we care especially about the error on the moments of the non-thermal subpopulation $M_{\mathrm{PIC}}$ to understand the ability of the respective methods to correctly capture the most dynamic subpopulation of particles. This is important because in many of the relevant applications of far-from-equilibrium, N-body systems we are interested in the spectrum of accelerated particles.

## 5 EXPERIMENTS

In this section, we aim to answer two questions: (1) does our LHPC model with hybrid representation achieve better accuracy than learning the evolution with a pure fluid representation? (2) Does LHPC offer speedup, compared to the full-PIC solver that evolves the full system in first principles with particle representation? We evaluate LHPC and baselines in the N-body laser-plasma interactions system described above. We evaluate both single-step prediction error and auto-regressive rollout error over 50 steps, where the state of the system changes significantly. We use relative L2 as the error metric – the ratio between L2 norm of error over L2 norm of ground-truth values.

### 5.1 DATA AND EXPERIMENT

Both the baseline and main pipeline are trained over 9 trajectories of PIC simulation. 10 trajectories were generated that vary over the normalized laser intensity $a_0$ from $[2, 4, \ldots, 20]$ and trajectory 5 ($a_0 = 10$) is held-out for testing (see Appendix A for details). In this way, we test the generalization of the models to novel initial conditions and environment setup. Each dataset consists of $T = 1500$ time steps and uses the separation strategy (details in Appendix B). For the nine datasets, we train with single-step loss on the time-range [0,1400], and validate on [1400, 1500]. We iterate on hyperparameters to obtain the best individual settings for both the baseline and hybrid pipelines respectively (detailed in Appendix). Then, the best models are finetuned using the multi-step loss with data from timeranges $[980, 1000] \cup [1080, 1100] \cup [1180, 1200] \cup [1280, 1300] \cup [1380, 1400]$, and validated on [1400, 1500]. We report the confidence intervals of the 1-, 20- and 50-step relative L2 errors on the held-out dataset and report the results in Table 1, which shows the error for the EM-field, the non-thermal particles ($M_{\mathrm{PIC}}$) and the full fluid, respectively. Runtime for ground-truth (GT) solver and the compared models are also provided in Table 1. We provide additional results in Appendix D with an ablation study on trajectory 10 with the most intense laser ($a_0 = 20$).

### 5.2 RESULTS

**Error comparison**. From Table 1, we see that our LHPC typically outperforms the baselines in terms of error by a wide margin. Among the three aspects we evaluate, the EM-field error and the non-thermal particles ($M_{\mathrm{PIC}}$) are the most important, since the EM-field is most sensitive to the local particle dynamics, and the non-thermal particles are typically the most important observable/deliverable of laser-plasma accelerators. An all-fluid description can be very efficient and accurate at describing the thermal ($M$) population, as is common for traditional fluid solvers. However, it fails to capture non-thermal particle acceleration, manifested in large errors for the field evolution (Field) and non-thermal population ($M_{\mathrm{PIC}}$). Specifically, for Field, our LHPC achieves an error reduction of 91.0%, 91.3% and 85.4% (6.8-fold) for 1-, 20- and 50-step autoregressive rollout, compared to the best models among the baselines. Similarly, in the evaluation of $M_{\mathrm{PIC}}$, our LHPC achieves an error reduction of 63.5%, 36.2%, 25.8% for 1-, 20- and 50-step predictions, respectively, demonstrating the ability of our LHPC to model accurately the dynamics of those energetic non-thermal particles. The Baseline CNN, although having the same architecture as our LHPC, cannot model accurately the non-thermal particles via either all-fluid or Bi-Gaussian representations, thus achieving much larger error. Specifically, while the use of a separate fluid description for the non-thermal population (Bi-Gaussian model) can help capture its impact on the system evolution, it still leads to significantly larger errors in the field evolution when compared to the LHPC model and cannot be used to fully describe the complex distribution of the non-thermal particles (see also Sec. 5.3). Comparing baseline and state-of-the-art FNO, we find that FNO performs comparable or slightly worse than Baseline CNN, likely due to the simpler architecture of CNN being more appropriate to capture the local fluid evolution. The LHPC (no-coupling) ablation shows that the coupling component is essential in ensuring good long-term accuracy, especially for Field and $M_{\mathrm{PIC}}$. On

| Method | Component | Error @ step 1 | Error @ step 20 | Error @ step 50 | Speed (s/step) |
|---|---|---|---|---|---|
| GT Solver (full PIC) | – | – | – | – | 9.21E-01 |
| FNO: All-fluid | Field | 5.77E-02 ± 1.52E-02 | 9.36E-01 ± 2.00E-01 | 2.57E+00 ± 8.56E-01 | 7.79E-02 |
| | $M_{\text{PIC}}$ | – | – | – | |
| | $M$ | 5.53E-03 ± 1.16E-03 | 6.79E-02 ± 1.47E-02 | 3.07E-01 ± 6.71E-02 | |
| FNO: Bi-Gaussian | Field | 2.98E-02 ± 2.36E-03 | 4.69E-01 ± 5.49E-02 | 9.65E-01 ± 1.51E-01 | 1.66E-01 |
| | $M_{\text{PIC}}$ | 3.25E-02 ± 8.10E-03 | 3.83E-01 ± 9.30E-02 | 1.03E+00 ± 4.75E-01 | |
| | $M$ | 6.92E-03 ± 1.17E-03 | 8.63E-02 ± 1.01E-02 | 2.38E-01 ± 3.45E-02 | |
| Baseline: All-fluid | Field | 1.97E-02 ± 7.53E-03 | 1.83E-01 ± 5.86E-02 | 7.16E-01 ± 3.66E-01 | **4.61E-02** |
| | $M_{\text{PIC}}$ | – | – | – | |
| | $M$ | 2.62E-03 ± 4.63E-04 | **4.86E-02** ± 9.02E-03 | **1.31E-01** ± 3.55E-02 | |
| Baseline: Bi-Gaussian | Field | 1.12E-02 ± 3.68E-03 | 1.31E-01 ± 2.61E-02 | 3.67E-01 ± 4.77E-02 | 1.10E-01 |
| | $M_{\text{PIC}}$ | 2.24E-02 ± 9.21E-03 | 1.82E-01 ± 7.70E-02 | 5.00E-01 ± 8.01E-02 | |
| | $M$ | 1.12E-02 ± 5.05E-04 | 6.07E-02 ± 8.69E-03 | 1.39E-01 ± 1.08E-02 | |
| LHPC (no-coupling) | Field | **1.01E-03** ± 1.72E-04 | 1.40E-02 ± 1.66E-02 | 1.38E-01 ± 4.52E-02 | 1.05E-01 |
| | $M_{\text{PIC}}$ | **8.18E-03** ± 2.02E-03 | 1.53E-01 ± 5.97E-02 | 5.15E-01 ± 1.15E-01 | |
| | $M$ | **4.51E-03** ± 7.12E-04 | 9.29E-02 ± 2.00E-02 | 3.26E-01 ± 7.00E-02 | |
| **LHPC** | Field | **1.01E-03** ± 1.57E-04 | **1.14E-02** ± 6.76E-04 | **5.34E-02** ± 6.68E-03 | 1.15E-01 |
| | $M_{\text{PIC}}$ | **8.18E-03** ± 2.02E-03 | **1.16E-01** ± 6.69E-02 | **3.71E-01** ± 7.83E-02 | |
| | $M$ | **4.51E-03** ± 7.12E-04 | 7.44E-02 ± 8.80E-03 | 1.80E-01 ± 2.88E-02 | |

Table 1: Results for the N-body laser-plasma interactions system on the held-out dataset. We evaluate three aspects of prediction: EM-field (Field), non-thermal particles ($M_{\text{PIC}}$), and the fluid moments $M$. Mean and 90% confidence interval are reported. The results are average performance on the held-out dataset over 15 independent rollouts with initial conditions at $t = 0, 100, \ldots, 1400$. Our LHPC achieves significant improvement on the EM-field predictions in both short- and long-terms. In terms of EM-field, LHPC achieves a 91.0%, 91.3%, and 85.4% (6.8-fold) reduction of error for 1-, 20- and 50-step rollouts, respectively, compared to the best-performing baseline. Similarly, for $M_{\text{PIC}}$ LHPC achieves error reduction of 63.5%, 36.2%, 25.8% for 1-, 20- and 50-step predictions. Note that the performance is N/A for all-fluid representation models for $M_{\text{PIC}}$ as they cannot model the non-thermal particles separately. For $M$ prediction, our LHPC has comparable performance with the baselines, since both use similar architectures to model the thermal part of the distribution as fluid, but the baselines also receive $M$ as an input whereas LHPC does not.

predicting the fluid moments ($M$), our LHPC model performs comparably to Baseline that has the same CNN architecture, showing both models can model reasonably the dynamics of the fluid.

**Runtime comparison**. While achieving significant error reduction on predicting the EM-field and the non-thermal particles, our LHPC also achieves significant speedup compared to the Ground-truth (GT) full PIC solver, with a reduction of runtime by 8.0-fold. We also see that the runtime of LHPC is not much higher than full deep learning based baselines, showing that although our LHPC involves a GT solver to evolve the non-thermal particles, the increase in runtime is negligible since the non-thermal particles only constitute a small fraction of the system.

## 5.3 Visualization of results

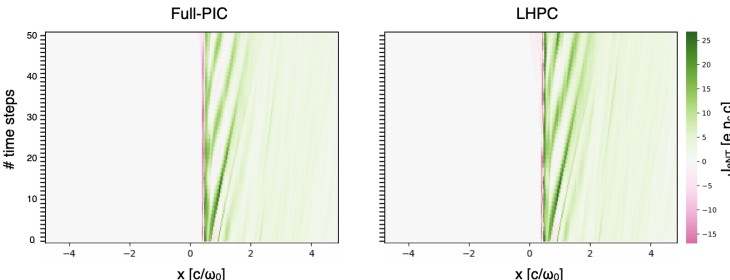

Figure 4: Comparison of the evolution of non-thermal electron current $J_{\text{eNT}}$ between ground truth (Full-PIC) and LHPC across 50 steps of rollout.

In this section, we visualize the predictions of LHPC. An accurate evolution of the system dynamics relies on the correct EM-field (Eq. 1g), which is dictated by the electric current $J$ (Eq. 1f). Figure 4

shows that our model can predict the evolution of $J$ contribution from the non-thermal population accurately over 50 time steps of rollout. This is further illustrated in Fig. 5, where for the different contributions of $J$, good agreement between the predictions of our model and the ground truth is observed. Similarly, our model predictions for components of the E-field and the thermal fluid density match well the ground truth after 20 steps of rollout. These results demonstrate faithful evolution of the important quantities that describe the dynamics of the fluid, the non-thermal particles, and the coupling between them. Importantly, our model is able to accurately capture the critical aspect of laser-plasma interactions — non-thermal particle acceleration (Fig. 6). Note that the highly non-trivial energy distribution of the accelerated (non-thermal) particles cannot be properly described with a few fluid moments and must rely on the accurate evolution of the particle representation.

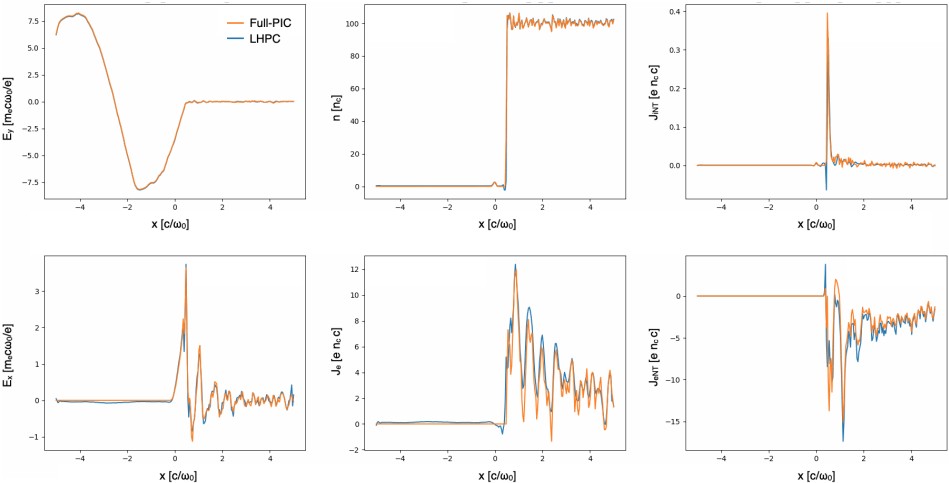

Figure 5: Comparison of ground truth (Full-PIC) and LHPC model predictions after 20 steps of rollout, for E-field components (left column) $E_x$, $E_y$, (middle) total density of the fluid $n$, and $J$ contribution from the fluid electrons $J_e$, (right) non-thermal ions $J_{iNT}$ and electrons $J_{eNT}$.

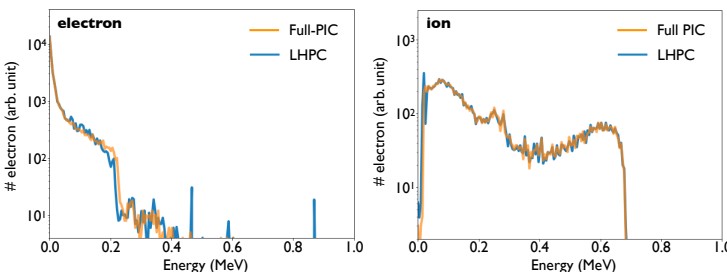

Figure 6: Comparison of energy spectrum of non-thermal electrons (left) and ions (right) between ground truth (Full-PIC) and LHPC after 20 steps of rollout.

## 6 DISCUSSION AND CONCLUSION

This work helps address a key challenge in modeling multi-scale, non-equilibrium, N-body systems by providing a novel hybrid particle-continuum model, whose coupling is learned from the data of first-principles simulations. LHPC outperforms the classical first-principles solver by an order of magnitude in speed, and the baseline method that models the entire system as a fluid by up to an order of magnitude in accuracy. Further speed-ups are expected in multiple dimensions as the non-thermal population constitutes a much smaller fraction of the system representation (see Appendix A.4).

Our method can be extended towards finding more efficient and accurate particle-continuum descriptions across different domains involving non-equilibrium N-body systems, and represents a key step towards the development of advanced data-driven algorithms for multi-scale modeling of nonlinear physical systems.

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

# Appendix

## A First-principles simulations of laser-plasma interactions

A plasma is a collection of unbound, moving ions and electrons, interacting through the electromagnetic (EM-) fields. It can be produced through for example the irradiation of a high-intensity ($I > 10^{18}$ W/cm$^2$) laser pulse onto a solid material such as a metal foil. Through the nonlinear interplay between the electrons, ions, and the EM-fields, these laser-plasma interactions can generate high-energy ($>$ MeV) ions for applications in material science (Patel et al., 2003), imaging (Rygg et al., 2008), and medical therapy (Kroll et al., 2022). To understand these nonlinear processes, it is necessary to use kinetic treatment of the plasma.

### A.1 Particle and continuum descriptions of plasmas

One example of kinetic treatment of plasma is the particle-in-cell (PIC) method (Dawson, 1983; Birdsall & Langdon, 1991). In this method we solve the Klimontovich equation (Klimontovich 1967) for finite size particles, coupled to Maxwell's equations for the EM-fields. The numerical procedure consists of solving Maxwell's equations on a spatial grid using the current and charge densities that are obtained by weighting the discrete plasma particles onto the grid. The particles are then advanced via the Lorentz force associated with the EM-fields. This particle-based simulation technique captures the kinetic microphysics of plasmas, and to the extent that quantum mechanical effects can be neglected, provides a first-principles description of plasma dynamics.

A plasma can also be described exactly as a continuum, by fluid equations – the evolution of the velocity moments of the distribution function $f$ of the plasma particles ($M_N(\mathbf{x}) := \int d^N v f(\mathbf{x}, \mathbf{v}) \mathbf{v}^N$ is the $N$-th order velocity moment). The fluid equations can be derived from the kinetic equations and form an infinite hierarchy of exact coupled conservation equations for each fluid moment (with the $n$-th moment depending explicitly on the $(n+1)$-th). In practice, this infinite hierarchy needs to be truncated after the first few moments, through the so-called closure relation – a relation that expresses the evolution of the highest-order moment considered in terms of the lower-order moments. For a plasma near local thermodynamic equilibrium, such as the *thermal* population of the electrons and ions studied in our work, these closures allow us to describe the plasma self-consistently at larger spatial and temporal scales (relative to the PIC scales), using a finite number of moments. Indeed, we use the first three moments – the mass, momentum, and energy densities of the particles – to describe the dynamics of the *thermal* fluids, and learn their coupling with the kinetic (*non-thermal*) particles and the EM-fields.

### A.2 Laser-plasma interactions

The generation of high-energy ion beams from intense laser-solid interactions has been an active area of research due to the potential of producing high-energy, high-charge, high-current ion beams in much more compact systems than solid-state based accelerators (*e.g.* linear accelerators, cyclotrons). The enormous accelerating gradients (TV/m; $\sim$5 orders of magnitude larger than solid-state based accelerators) can be sustained in a plasma (Wilks et al., 2001), accelerating ions to MeV energies within millimeters.

For these laser intensities it is useful to define the normalized vector potential $a_0 \simeq 0.85\sqrt{I[\text{W/cm}^2](\lambda_0[\mu\text{m}])^2/10^{18}}$, where $\lambda_0$ is the laser wavelength. The laser electric field can accelerate electrons to relativistic speeds in one cycle if $a_0 > 1$. For ion acceleration, a solid-density target is typically used, for which the laser E-field accelerates the electrons in the small, skin depth, layer near the front surface, producing very energetic electrons. In addition to the acceleration near the front surface, these energetic electrons cross the dense target and escape into the vacuum on the rear side, setting up a strong charge separation E-field that accelerates the ions from the back surface. As a result, the ions will be laminar and possess small divergence angle, and exhibit an energy spectrum typically characterized by an exponentially decreasing distribution (Snavely et al., 2000; Wilks et al., 2001).

### A.3 Simulation parameters

In our PIC simulations we consider an intense laser interacting with a planar, solid-density target. In our 1D simulations a laser with frequency $\omega_0$ is launched along the $x$ direction from the left boundary and irradiates an electron-proton plasma (*i.e.* $m_i = m_p = 1836\, m_e$). The laser pulse duration is $\simeq 15$ fs, with intensity $2 \leqslant a_0 \leqslant 20$. The plasma density follows a step-like profile with thickness $\simeq 5\,\mu$m and electron number density $10^{23}$ cm$^{-3}$ (considering a laser wavelength of $1\mu$m), corresponding to a solid-density target such as a metal foil.

The target is simulated with 1000 particles per cell per species, and the total simulation domain of $\simeq 80\,\mu$m is resolved with a spatial resolution (cell size) of $0.03\, c/\omega_0$. The time step is chosen according to the Courant–Friedrichs–Lewy condition, and the system is evolved over 2000 time steps (or 8500 for $a_0 = 20$). Periodic boundary conditions for both particles and fields are used.

### A.4 Computational challenges and opportunities

Studying these laser-plasma interactions using the particle-in-cell (PIC) method requires resolving the fastest and smallest oscillations of the electrons in the plasmas. As a consequence, modeling these interactions with first-principles simulations is very computational demanding. For example, a 3D one-to-one simulation of laser-plasma interactions typically involves evolving the dynamics of billions of (numerical) particles on a billion-cell grid, requiring millions of CPU hours to compute. In this work, we consider only one spatial dimension $x$, while retaining the three dimensions in momentum. This is identical to simulating a particle distribution uniform in the $y$ and $z$ directions. See Appendix A for details of laser-plasma interactions and the PIC method.

It is important to note that in more realistic 2D and 3D geometries a significant enhancement of speed-up can be achieved, due to the much smaller proportion of the non-thermal particles. This is a result of the finite spot size of the laser, which will only interact with a finite volume of the plasma, and accelerate a small fraction of the particles to non-thermal. In a 3D simulation of these laser-plasma interactions, the non-thermal particles compose typically $< 0.1\%$ of the total number of particles while encompassing $> 50\%$ of the system energy. Compared with the 1D simulations presented in this work ($\simeq 20\%$), one expects a speed-up of $> 1000$x in 3D.

## B Details for separation method

Here we describe the details of the separation method for preparing the labeled (thermal and non-thermal) data. The algorithm separates the particles (from the original simulations) into thermal and non-thermal populations. This is done by computing locally the moments of the particle velocity distribution $f(v)$ and considering non-thermal particles those with velocities $v$ that exceed a threshold $\alpha v_{\text{th}}$ – a given multiple ($\alpha$) of the thermal velocity $v_{\text{th}}$ (= the velocity spread $\sigma_v$ for non-relativistic velocities; Cohen et al. 2010; Fiuza et al. 2011). At each time step, we study the velocity distribution function of the particles in the local neighborhood in space $\mathcal{N}_i$ of each particle $\mathcal{P}_i$. Without knowing which particles belong to the thermal population *a priori*, we begin by identifying a population that is likely thermal at iteration 1. The velocity spread $\sigma_{v_1}$ computed from this population provides a first estimate of the threshold $\alpha\sigma_v$, allowing us remove particles with $v > \alpha\sigma_v$ from the distribution. The procedure is repeated (on the updated distribution) until a convergence of the value of the threshold is reached at iteration $N$. The remaining particles now constitute the thermal population, characterized by the fluid velocity $v_{\text{fl}} = \langle v_j \rangle \, \forall\, j \in \mathcal{N}$ and thermal velocity $v_{\text{th}} = \sigma_{v_N}$ ($\langle \cdots \rangle$ denotes the average value). These are the mean and sigma of the Gaussian. The corresponding threshold $\alpha v_{\text{th}}$ is then used to determine the population $\mathcal{P}_i$ belongs to. Note that only $\mathcal{P}_i$ has been given a label, the other particles in $\mathcal{N}_i$ were only used to calculate $v_{\text{fl}}$ and $v_{\text{th}}$.

Figure 7 illustrates an iteration in detail. From the example velocity distribution function shown in Fig. 7(a), we recognize an absolute peak located at $v_{\text{abs}}$ with height $h_{\text{abs}}$. Among the peaks $(v_\mu, h_\mu) \, \forall\, h_\mu \leqslant X h_{\text{abs}}$ for a constant $X$, pick the one with $v_\mu$ closest to zero as the first guess of thermal population and denote the peak location as $(v_0, h_0)$ (Fig. 7b). Search down and outwards from $(v_0, h_0)$ (*i.e.* in both directions of $v$ with $v_+ > v_0 > v_-$) until the heights $h_+ \equiv h(v = v_+) = Y h_0$ and $h_- \equiv h(v = v_-) = Y h_0$ (Fig. 7c). The population enclosed by $\{v | v_+ > v > v_-\}$ is regarded as the population to estimate the threshold. Transform all the particles in $\mathcal{N}$ into the local

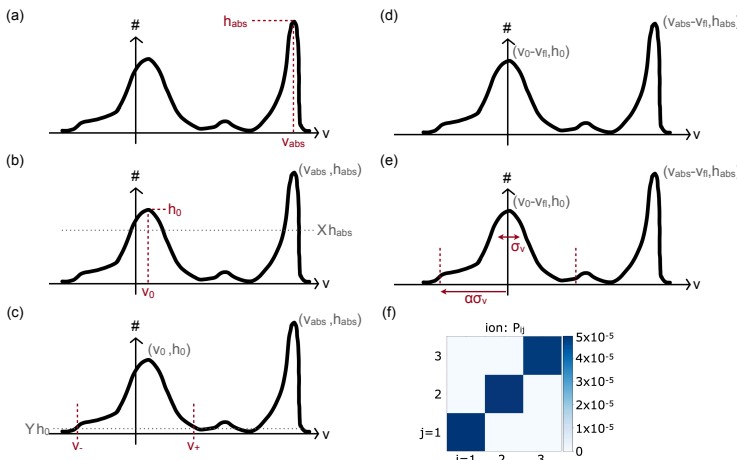

Figure 7: The separation method identifies the thermal and non-thermal populations from the local velocity distribution of particles.

fluid frame according to $v_{\text{fl}}$ (Fig. 7d; for non-relativistic velocities this amounts to $v_i \rightarrow v_i - v_{\text{fl}}$). Calculate $\sigma_v$ to obtain the threshold (Fig. 7e).

The size of $\mathcal{N}_i$, values of $X$, $Y$, and $\alpha$ are tunable parameters determined by the physics and fine-tuned empirically. We have found that $\mathcal{N}_i \simeq 60$ cells, $X = 0.8$, $Y = 0.1$, and $\alpha = 3$ and 5 for the water and laser-plasma systems lead to the most physically accurate separation.

The accuracy of the algorithm is evaluated by the pressure tensor $P_{ij}$ of the thermal population, where $P_{ij} = \langle v_i v_j \rangle - \langle v_i \rangle \langle v_j \rangle$ characterizes the second-order velocity moments and a is directly related to how well the distribution is described by a Gaussian. Figure 7(f) exemplifies that indeed, for the identified thermal population of the ions for the laser-plasma system, the diagonal terms are all comparable and the off-diagonal terms are negligibly small. Namely, the velocity distribution can be described by a Gaussian.

## C    FULL PIPELINE

### C.1    COMPONENT DETAILS

#### C.1.1    COMPONENTS (A) AND (B)

For both $M$ or $M_{\text{NT}}$, there's 20 moment directional components that needs to be predicted - 10 for ions, 10 for electrons - of the zero (1), first (3), and second (6) moment orders. Thus, for both components, we instantiate 20 models with the same hyperparameter configuration in $f_M$ or $f_{M_{\text{NT}}}$. We also train Component (1) with the push-forward trick and multi-step loss, shown in Brandstetter et al. (2022) to greatly enhance capability of generalization.

#### C.1.2    COMPONENT (C)

In order to parameterize the distribution to sample, a three-dimensional Gaussian is taken to have mean $\mathbb{E}[u] \approx$ the first order moments in $\hat{M}_{\text{NT}}^{t+1}$, and covariance matrix $\mathbb{E}[(u - \mathbb{E}[u])(u - \mathbb{E}[u])^T] \in \mathbb{R}^{3 \times 3}$ informed by the second order moments in $\hat{M}_{\text{NT}}^{t+1}$. To be able to sample, we require a valid (positive definite) covariance matrix. In order to enforce this condition, we obtain the ground-truth $L$ using the Cholesky decomposition $\mathbb{E}[(u - \mathbb{E}[u])(u - \mathbb{E}[u])^T] = LL^T$ and train $f_{M_{\text{NT}}}$ to predict $\hat{L}$. During rollout, we approximate $\mathbb{E}[(u - \mathbb{E}[u])(u - \mathbb{E}[u])^T] \approx \hat{L}\hat{L}^T$, and use the valid covariance matrix to sample a set of particles which is then "injected". The cumulative set of injected particles constitute the PIC loop.

### C.1.3 COMPONENT (D)

After obtaining the sample of injected particles, we subtract their moments from $\hat{M}^{t+1}$ to obtain the thermal population's moments $\hat{M}'^{t+1}$ for the next time step.

### C.1.4 COMPONENTS (E) (F) (G)

Each of these components by default use the respective components from the imitation solver. The imitation solver implements the equations used in the OSIRIS PIC simulation, and is tested to match the simulation to machine precision error on any single timestep. These components run efficiently because we now only advance the much smaller sub-population of non-thermal particles $(x, \mathbf{u})^{t+1}$. Figure 8 illustrates the full pipeline of our method, concretized into the laser-plasma simulation.

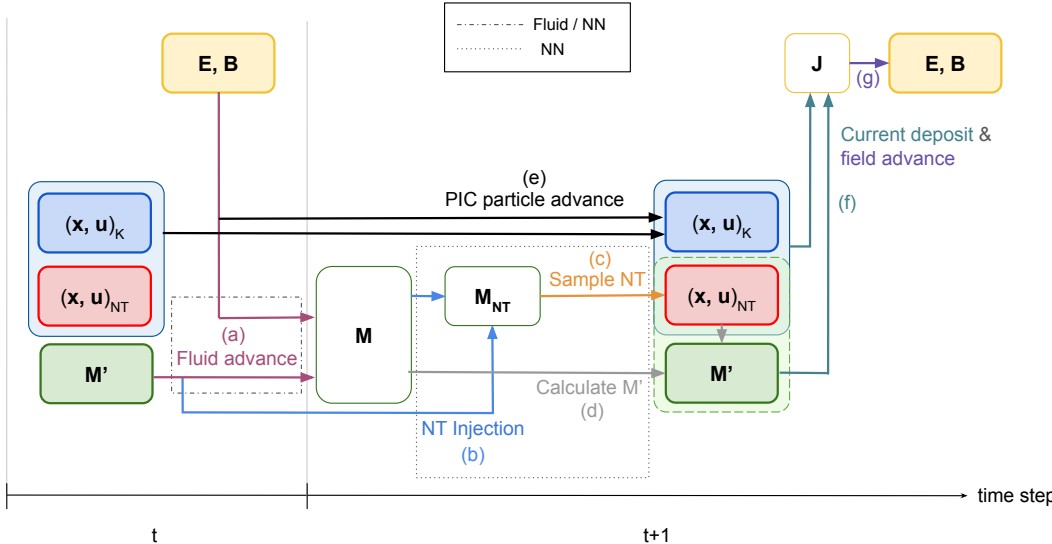

Figure 8: Schematic of LHPC for the multi-scale laser-plasma interaction, which is the application of the general pipeline (Fig. 1) to the current problem. The components correspond almost exactly to Eqs. 1a - 1g except current deposit and field advance are separated.

## D ADDITIONAL RESULTS FOR MULTI-SCALE LASER-PLASMA INTERACTION

Here we provide additional results for our experiments, as shown in Tables 2 and 3. Table 2 uses the same dataset split as Table 1, but focus on rollout starting from $t = 1200$, where there is most injection and dynamical activity. From the table, we see that the conclusion is similar as Table 1 in main text: LHPC achieves significant error reduction for key quantities of interest: EM field and MPIC (energetic particles).

To ablate different aspect of our model, we perform an additional experiment with a different train: test split, where the test dataset is in the *future* of the training (Table 3). We take the 10[th] trajectory with the highest laser intensity that consists of $T = 8480$ time steps, where the training time-range is [1000, 1900] and the testing (rollout) time-range is 1900 onwards. We explore an ablation study on various strategies to use the additional data [1800, 1900]. Namely, we compare finetuning with simply retraining the model on [1000, 1900]. We also use the multi-step loss with the push-forward trick, reported by Brandstetter et al. (2022) and other works to enhance the generalization of the model, and because it can accurately mimic the input distribution during rollout.

## E ADDITIONAL DETAILS FOR ARCHITECTURE AND TRAINING

We use a convolutional network (CNN) as the base architecture for modeling fluid network $g_{\text{fluid},\theta}$ (Fig. 9) and injection network $g_{\text{inject},\varphi}$. For both networks, we use the same 6 layers, with the same

| Method | Component | Error @ step 1 | Error @ step 10 | Error @ step 20 | Error @ step 50 | Speed (s/step) |
|---|---|---|---|---|---|---|
| GT Solver (full PIC) | – | – | – | – | – | 9.21E-01 |
| FNO: All-fluid | Field | 4.02E-02 | 3.45E-01 | 6.45E-01 | 1.28E+00 | |
| | $M_{\text{PIC}}$ | – | – | – | – | 8.45E-02 |
| | $M$ | 8.91E-03 | 6.64E-02 | 9.58E-02 | 4.20E-01 | |
| FNO: Bi-Gaussian | Field | 3.17E-02 | 2.49E-01 | 4.00E-01 | 3.80E-01 | |
| | $M_{\text{PIC}}$ | 3.16E-02 | 1.31E-01 | 2.17E-01 | **4.40E-01** | 1.69E-01 |
| | $M$ | 9.12E-03 | 5.85E-02 | 9.88E-02 | 2.98E-01 | |
| Baseline: All-fluid | Field | 7.39E-03 | 4.27E-02 | 9.05E-02 | 2.71E-01 | |
| | $M_{\text{PIC}}$ | – | – | – | – | **4.88E-02** |
| | $M$ | 5.23E-03 | **3.16E-02** | **6.62E-02** | **1.46E-01** | |
| Baseline: Bi-Gaussian | Field | 6.62E-03 | 4.63E-02 | 1.09E-01 | 2.98E-01 | |
| | $M_{\text{PIC}}$ | 1.53E-02 | 4.95E-02 | 8.66E-02 | 4.52E-01 | 1.27E-01 |
| | $M$ | **5.22E-03** | 3.50E-02 | 7.18E-02 | 1.62E-01 | |
| LHPC (no-coupling) | Field | **1.10E-03** | 9.27E-03 | 1.39E-02 | **7.32E-02** | |
| | $M_{\text{PIC}}$ | **1.02E-02** | 4.16E-02 | 7.58E-02 | 1.92E-01 | 1.05E-01 |
| | $M$ | 6.22E-03 | 7.05E-02 | 1.48E-02 | 3.80E-01 | |
| **LHPC** | Field | **1.10E-03** | **8.08E-03** | **1.07E-02** | 7.33E-02 | |
| | $M_{\text{PIC}}$ | **1.02E-02** | **1.89E-02** | **3.70E-02** | 6.62E-01 | 1.15E-01 |
| | $M$ | 6.22E-03 | 4.31E-02 | 9.65E-02 | 2.86E-01 | |

Table 2: Results for laser-plasma interactions for time range $t = 1200$–$1250$. Trajectory 5 ($a_0 = 10$) is held out for testing, and the model is trained on the other 9 datasets. We report performance on Dataset 5 and rollout at $t = 1200$, where there is the most injection and dynamical activity. As shown, the model remains stable within this chaotic, unseen time range, showing generalization to an unseen trajectory.

| Method | Trained on | Finetuned on | Finetuning multi-step loss | Rollout L2 step 1 | Rollout L2 step 5 | Rollout L2 step 10 | Rollout L2 step 20 | Rollout L2 step 30 | Speed (s/step) |
|---|---|---|---|---|---|---|---|---|---|
| Ground-truth Solver (full PIC) | N/A | N/A | N/A | (N/A, N/A) | (N/A, N/A) | (N/A, N/A) | (N/A, N/A) | (N/A, N/A) | 1.71E+00 |
| Baseline: All-fluid | 1000-1800 | N/A | N/A | (9.19E-02, 6.22E-03) | (3.81E-01, 2.07E-02) | (8.49E-01, 4.48E-02) | (2.16E-01, 9.31E-02) | (3.94E-01, 1.46E-01) | 2.31E-02 |
| | 1000-1900 | N/A | N/A | (2.68E-02, 5.10E-03) | (6.53E-02, 1.87E-02) | (1.08E-01, 4.01E-02) | (2.27E-01, 7.83E-02) | (3.80E-01, 1.46E-01) | 2.58E-02 |
| | 1000-1800 | 1800-1900 | 1 | (1.73E-02, 5.13E-03) | (5.17E-02, 2.22E-02) | (8.99E-02, 4.28E-02) | (5.18E-01, 1.48E-01) | (5.28E-01, 7.45E-01) | 2.20E-02 |
| | 1000-1800 | 1800-1900 | 1:1 2: 0.5 3:0.1 | (1.68E-02, 4.88E-03) | (4.79E-02, 2.13E-02) | (7.66E-02, 3.99E-02) | (1.72E-01, 8.21E-02) | (2.53E-01, 1.25E-01) | **2.13E-02** |
| LHPC | 1000-1800 | N/A | N/A | (3.78E-03, 4.44E-03) | (**1.01E-02**, 2.74E-02) | (1.80E-02, 3.84E-02) | (5.98E-02, 9.48E-02) | (1.21E-01, 2.06E-01) | 1.80E-01 |
| | 1000-1900 | N/A | N/A | (**3.73E-03, 4.31E-03**) | (1.09E-02, 1.99E-02) | (2.12E-02, 3.96E-02) | (4.41E-02, 9.81E-02) | (8.42E-02, 2.40E-01) | 1.76E-01 |
| | 1000-1800 | 1800-1900 | 1 | (3.78E-03, 5.34E-03) | (1.10E-02, **1.47E-02**) | (**1.88E-02, 3.41E-02**) | (6.92E-02, 1.02E-01) | (2.08E-01, 2.30E-01) | 1.78E-01 |
| | 1000-1800 | 1800-1900 | 1:1 2:0.5 3:0.1 4:0.1 | (4.32E-03, 6.46E-03) | (2.57E-02, 2.07E-02) | (4.02E-02, 3.60E-02) | (**4.15E-02, 7.11E-02**) | (**7.10E-02, 1.19E-01**) | 1.64E-01 |

Table 3: Ablation study for multi-physics laser-plasma interactions for the 10th dataset that has highest laser intensity ($a_0 = 20$). We compare amongst: (a) train only single-step model from [1000, 1800] vs. (b) from [1000, 1900] vs. (c) taking (a) and finetuning with single-step loss on [1800, 1900] vs (d) taking (a) and finetuning with multi-step push-forward trick loss on [1800, 1900]. We compare with the All-fluid baseline, and report result from the best hyperparameter configuration. Each cell reports the relative L2 error for the EM-field $[E^t, B^t]$, for $M^t$.

kernel sizes of 1,1,3,7,3,1, with feature sizes of 64, allowing feature extraction and local information exchange among neighboring cells. For each feature of the moments, we use a different feature head as shown in Fig. 9.

**Training**. We use Adam (Kingma & Ba, 2014) optimizer, with starting learning rate of $10^{-3}$. The training consists of two stages. In the first stage, we train both $g_{M,\theta}$ and $g_{M_{NT},\varphi}$ with single-step loss:

$$L_1 = \mathbb{E}_t \left[ \ell(\hat{M}^{t+1}, M^{t+1}) + \ell(\hat{M}_{\text{NT}}^{t+1}, M_{\text{NT}}^{t+1}) \right] \tag{2}$$

In the second stage, we fine-tune with predicting $N = 4$ steps into the future with push-forward trick (Brandstetter et al. 2022 that stops the gradient on the input. The loss is given by:

$$L_2 = \mathbb{E}_t \left[ \sum_{i=1}^{N} \alpha_i \ell(\hat{M}^{t+i}, M^{t+i}) + \sum_{i=1}^{N} \alpha_i \ell(\hat{M}_{\text{NT}}^{t+i}, M_{\text{NT}}^{t+i}) \right] \tag{3}$$

For both losses, we have

$$\hat{M}^{t+i} = g_{M,\theta}\left(\mathrm{sg}((\hat{E},\hat{B})^{t+i-1}), \mathrm{sg}(\hat{M}'^{t+i-1})\right) , i = 1, 2, ...N \tag{4}$$

$$\hat{M}_{\mathrm{NT}}^{t+i} = g_{M_{\mathrm{NT}},\varphi}\left(\mathrm{sg}(\hat{M}^{t+i-1}), \mathrm{sg}(\hat{M}'^{t+i-1})\right) , i = 1, 2, ...N \tag{5}$$

Notice the hat notation $\hat{\cdot}$ in the input arguments, denoting that they are autoregressive prediction of the LHPC at the previous time step, but the gradient is stopped (the "sg" notation). Essentially, we rollout the pipeline of LHPC for multiple steps, and use it to provide a realistic input that contains the rollout error, requiring the model to not only predict well, but able to adapt to noise due to the rollout error. We find that this significantly improve the long-term prediction performance. For the coefficient of the multi-step loss in Eq. 4, we set $(\alpha_1, \alpha_2, \alpha_3, \alpha_4) = (1, 0.5, 0.1, 0.1)$ with decreasing weight for longer time steps. This put more emphasis on the single-step loss to make the training more stable, and also have weight on longer-term future to improve long-term prediction.

For the loss function $\ell(\cdot, \cdot)$ in Eq. 3 and 4, we use

$$\ell(\hat{y}, y) = |\hat{y} - y|^{1.5} \tag{6}$$

We find that this achieves a better performance than the alternative of MSE loss (with exponent of 2) and MAE loss (with exponent of 1). MSE will give very small gradient if the loss is small, and not able to encourage that the prediction to be exactly 0 in the vacuum. On the other hand, MAE is harder to train and do not penalize more for larger errors. Our choice of loss function $\ell(\cdot, \cdot)$ strikes a good balance between the two and enjoys the benefit of both loss functions.

For single-step and multi-step training, we both train 500 epochs, with and cosine learning rate scheduling (Loshchilov & Hutter, 2016).

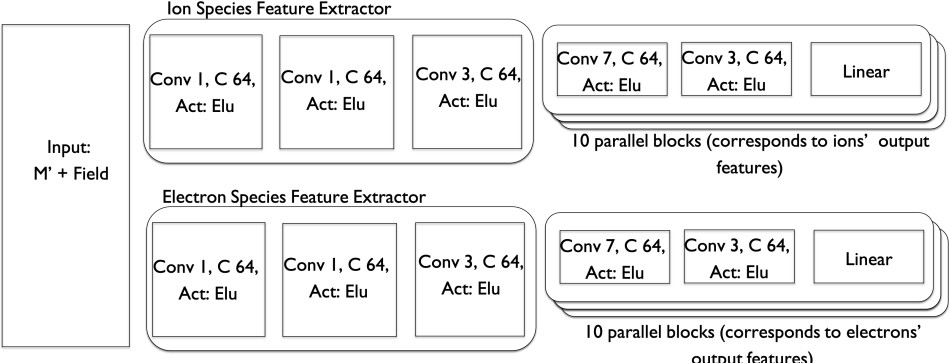

Figure 9: Schematic of the architecture of $g_{\mathrm{fluid},\theta}$.

