# OpenReview forum: "Learning Efficient Hybrid Particle-continuum Representations of Non-equilibrium N-body Systems"
_ICLR.cc/2023/Conference — Submitted to ICLR 2023_

### Official Review · Reviewer_qT1Y · 2022-10-15

**Confidence:** 3
**Correctness:** 3
**Technical Novelty And Significance:** 2
**Empirical Novelty And Significance:** 3
**Recommendation:** 5

**Clarity, Quality, Novelty And Reproducibility:**

Overall, this paper is well written. The idea and model details are clear enough.
Beyond the uniform representation, using hybrid representation achieves good performance on nonlinear physical systems.
It is not surprising and it is better to provide more analysis of different information coupling methods.


For reproducibility, the details of the neural networks are attached in the supplementary material. But it is better to provide the source code so that more details can be checked.

**Strength And Weaknesses:**

Strength:
1）This paper focuses on an important and interesting problem that models the non-equilibrium, N-body nonlinear， which has the potential for complex physical systems analysis.
2) The hybrid representation makes sense and works well in the simulations.
3) The basic idea and technical details are clear.  More details of the network architecture are attached in the supplementary details.

Weakness:
The main weaknesses of this paper lie in experimental comparison and technical contribution:
1）There seem to be some existing methods in the field of nonlinear physical systems modeling, such as LED[1] and  Solver-in-the-loop[2]
It is suggested to compare the proposed method with these existing methods.
2)  There are some methods to fuse the representation of different views, such as Cross-Attention and CCA. Do these information-fusing methods work well in physical systems modeling?
3) The whole method contains three components.  Which one is most important? It is suggested to add a detailed ablation study.
4) Using a neural network to learn particle-continuum coupling is not difficult. Did existing methods use the neural network for coupling? If yes, which is the advantage of this method? If not, it is better to analyze the effect of different deep-learning-based coupling methods.




**Summary Of The Paper:**

This paper proposes to jointly formulate particle and continuum for model multi-scale, non-equilibrium, and no-body physical systems. Specifically,  this model contains three components, 1) a classical solver for kinetic particles, 2) a neural network for the evolution of thermal sub-population with fluid representation, and 3) an injection neural network from a fluid state to a particle state.

Compared with existing methods on non-equilibrium, and no-body physical systems, it uses the hybrid representation instead of the uniform representation. Though the method is simple, it achieves a good balance between accuracy and speed, i.e. faster than classical particle solvers and more accurate than the deep-learning baselines.



**Summary Of The Review:**

Balancing the strength and weaknesses, I think this paper would be better when more comparisons and analyses are conducted.
Besides, the technical contribution of adding a neural network to learn particle-continuum coupling is limited.

---

> ### Author Response · Authors · 2022-11-16
> **Official Response (3)**
>
> > There are some methods to fuse the representation of different views, such as Cross-Attention and CCA. Do these information-fusing methods work well in physical systems modeling?
>
> Answer:  While these approaches can be useful for modeling some physical systems, they are not appropriate for the wide class of out-of-equilibrium N-body systems that we address in our work. Both cross-attention and CCA require that both modalities are fixed size vectors, which in our case, would require that both modalities be fluid. As we discussed above, this would not be appropriate to the physical systems of interest, where the two modalities are one fluid and one changing number of particles, respectively. Thus, standard information-fusing methods such as cross-attention and CCA cannot be directly applied here. Moreover, CCA uses linear projection, which is not appropriate to our class of problems where the coupling is highly nonlinear (e.g., the injection of particles from fluid to particle states depends on a complex interplay between the EM field and the fluid moments).
>
> > Using a neural network to learn particle-continuum coupling is not difficult. Did existing methods use the neural network for coupling? If yes, which is the advantage of this method? If not, it is better to analyze the effect of different deep-learning-based coupling methods.
>
> Answer: We respectfully disagree that using a neural network (NN) to learn particle-continuum coupling is not difficult. For systems that are far from equilibrium (i.e. when the particle distributions deviate strongly from Gaussian distributions) this relation is far from trivial. Moreover, the physics of coupling in most problems is highly nonlinear and not well understood, making it challenging to understand what is the most appropriate NN architecture to use and to ensure stability, physical consistency and generalizability of the method.
>
> To the best of our knowledge, there are no established methods that use neural networks for coupling particle and continuum representations. Previous works on the use of machine learning to improve simulation modeling of nonlinear physical systems either used uniform particle representations or uniform fluid representations. As we discussed in the previous point, in systems near-equilibrium where a uniform representation may be satisfactory, information-fusing methods like cross-attention or CCA can be advantageous. However, in far-from-equilibrium, N-body systems this is not sufficient as shown by the complex particle distribution of Figure 6 (which is far from Gaussian) and by the comparison of our results with “Baseline: Bi-Gaussian” and “FNO: Bi-Gaussian” ablations, where we use another fluid representation to model the far-from-equilibrium particles. From Table 1 and its analysis, we see that these lead to much larger errors for far-from-equilibrium particles, which are typically the quantity of interest for many of the applications (the spectrum of the accelerated particles). New approaches must be developed to couple the hybrid particle-continuum representations needed for far-from-equilibrium N-body systems.
>
> To the best of our knowledge, our work represents the first time ML methods are being used to couple particle and continuum descriptions to efficiently model a broad class of far-from-equilibrium N-body systems. The learned particle-continuum coupling must capture how mass, momentum and energy from the near-equilibrium fluid (continuum) representation is transferred (injected) to the nonthermal (non-equilibrium) particle representation. Our method follows what we believe is the most natural and robust way to learn this exchange: we first predict the moments of the distribution of non-thermal particles to be injected at any given time and location from the near-equilibrium fluid component. We then sample a population of particles from this distribution. And finally, this injected population is evolved with the ground truth particle solver providing an accurate description of the most important part of the distribution but with a very large overall speedup on the computation of the system evolution. Our results demonstrate that this new approach is appropriate and effective by maintaining good accuracy while significantly speeding up the calculations. This is a very important contribution in Machine Learning for Sciences that we are confident will be of interest to the ICLR community.
>
>
> > For reproducibility, the details of the neural networks are attached in the supplementary material. But it is better to provide the source code so that more details can be checked.
>
> Answer: Thanks for raising this point. We will open-source the data and the code upon publication of the paper.

---

> > ### Comment · Reviewer_qT1Y · 2022-11-19
> > **Quick response to authors**
> >
> > Dear Authors,
> >
> > I appreciate the author's detailed point-to-point responses.  I will read them carefully and give feedback later.
> >
> > Many thanks,
> >
> > Reviewer qT1Y

---

> > ### Comment · Reviewer_qT1Y · 2022-12-10
> > **Response to authors**
> >
> > Thanks very much for the authors' detailed responses.
> >
> > My concerns on model comparisons with other baseline methods and self-ablations are well addressed.
> >
> > For the information-fusing methods, I still think they may work for the N-body systems though both modalities are fixed-size vectors.  For example, the information can be fused in the latent space instead of the data level.  Besides, both the attention models and the cca methods can be extended to non-linear cases.
> >
> > I am not convinced that using a neural network to obtain a nonlinear representation is difficult.  If it is challenging to understand what is the most appropriate NN architecture to use, it is necessary to conduct evaluations for different NNs and provide the analysis.  What are the challenges in model building and how are they overcome? What are the advantages of the proposed one over the others?
> >
> > Based on partially addressed concerns, I will keep my score unchanged.

---

> > > ### Author Response · Authors · 2022-12-12
> > > **Response**
> > >
> > > We thank the reviewer for the comments.
> > >
> > > As explained in the introduction of our paper, the development of efficient hybrid particle-continuum solvers has been restricted by the challenge in establishing an accurate coupling between the particle and continuum representations. Indeed, the standard approach in the literature is to postulate an ad hoc model that is mostly only qualitatively anchored in our incomplete understanding of the physics. We have therefore relied on the first principles simulations in the original space (data level) that are able to capture the coupling physics accurately, while achieving the important improvements in speed as demonstrated in our work.
> > >
> > > Specifically, we did not consider methods based on fusing the particle and continuum representations at the latent level because of the following reasons:
> > >
> > > 1. The coupling model would not be physically interpretable regarding the exchange of physical quantities such as mass and momentum between the particle and continuum populations.
> > > 2. Precision is critical in many of the domains of interest, for which representing the non-thermal (particle) population using a fixed-sized vector would be limited. In contrast, our method can expand the particle subset to control how much precision is required.
> > > 3. Conservation laws and other physically-informed properties are also important for the accurate understanding of the physics involved in the evolution and the coupling of the two representations. However, enforcing those laws and properties in methods based on dynamics at the latent space level have been shown to rely on explicit knowledge of the underlying equations, which is fundamentally lacking in our case.
> > >
> > > Again, we thank the reviewer for the suggestions, and we agree that conducting evaluations with different NNs is important. Indeed, we have done this in our work and we have shown the importance of the strategies developed:
> > >
> > > 1. We have done extensive experiments on NNs to predict the coupling (MNT) and evolution of the thermal component (M), trying different combinations of kernel sizes, number of layers, branching, loss function, etc. to identify the most accurate options. This work also revealed that using a NN architecture based on the structure of the velocity moment (fluid) equations led to improved accuracy. We realize that this part could have been better documented in the paper and we can add a table and discussion of the different tests in the camera ready version.
> > > 2. The multi-step training of the NNs was critical to ensure long-term stability of the algorithm as can be seen in Table 3 of the paper.
> > > 3. The NNs developed are demonstrated to outperform the state-of-the-art deep learning-based surrogate model of Fourier Neural Operator (FNO) as can be seen in Tables 1 and 2 of the paper.
> > >
> > > We agree that further analysis can be done and will be done in future work. However, the current analysis clearly supports the claims of our work, demonstrating that the proposed method achieves an important balance between accuracy and speed, compared with classical first-principles particle solvers and deep-learning baselines with uniform representations, which is the main focus of this paper.

---

> ### Author Response · Authors · 2022-11-16
> **Official Response (2)**
>
> > The whole method contains three components. Which one is most important? It is suggested to add a detailed ablation study.
>
> Answer: Thank you for this important suggestion The three components of LHPC are: (1) a neural network evolving the near-equilibrium particles with fluid representation; (2) a ground-truth solver that evolves the far-from-equilibrium particles with particle representation; (3) a neural network that models the coupling between the two populations. Any reasonable ablation needs to still model the complete state of the system. For example, if we ablate the component (1), i.e. without any fluid representation, then the component (2) ground-truth solver needs to evolve the full system as particles, and cannot only evolve parts of the system (otherwise the description of the system is incomplete). Similarly, if we ablate component (2), then the full system needs to be represented as a fluid. Based on this principle, in Table 1 and Table 2 (in Appendix D) of the original submission, we provided the results of different ablations of our model, which include:
>
> * (a) Using an all-particle representation, and evolving it with the ground-truth solver (“GT solver (full PIC)”). This is ablating component (1).
> * (b) Using an all-fluid representation, with CNN or FNO as the base architecture (“Baseline: All-fluid” and “FNO: All-fluid”). This is ablating component (2), and putting this subset of far-from-equilibrium particles into fluid representation.
> * (c) Using a fluid representation for the near-equilibrium particles, and using another fluid to represent the far-from-equilibrium particles, with CNN or FNO as the base architecture (“Baseline: Bi-Gaussian” and “FNO: Bi-Gaussian”). This is ablating component (2), and uses another fluid to represent this subset of far-from-equilibrium particles.
>
> In the updated draft, and following the reviewer’s suggestion, we have also **added** one more ablation into Table 1 and 2:
> * (d) Using both fluid and particle representations, but without the coupling component (“LHPC (no coupling)”). This is ablating component (3).
>
> The detailed analysis of different ablations is in the Sec 5.2 of the main text. A summary is as follows: from ablation (a), we see that:
>
> * Component (1) of representing the near-thermal part of the distribution (that contains most of the particles) as fluid is critical for accelerating the runtime (8x speedup)
> * Component (2) of representing the non-thermal parts of the system is critical for significantly reducing the error of key quantities of interest (Field and M_PIC) (up to 6.8-fold error reduction than without)
> * Component (3) of having the coupling between the two representations is also critical for significantly reducing the error of key quantities of interest (Field and M_PIC) (up to 2.6-fold error reduction than without).
>
> Therefore, all the three components of LHPC are essential in ensuring a faster and more accurate long-term prediction of this multi-scale, multi-physics N-body system.

---

> ### Author Response · Authors · 2022-11-16
> **Official Response (1)**
>
> We thank the reviewer for the thoughtful comments. Below we address the reviewer’s points.
>
> > There seem to be some existing methods in the field of nonlinear physical systems modeling, such as LED[1] and Solver-in-the-loop[2] It is suggested to compare the proposed method with these existing methods.
>
> Answer: Indeed both LED and solver-in-the-loop also integrate conventional numerical solvers with neural networks, but are based on uniform (fluid) representations, which are not appropriate to model N-body systems far from equilibrium as shown in our work. These approaches address a complementary problem to ours.
>
> More specifically, LED is a powerful approach for multi-scale fluid simulations, where it optimizes the **temporal** dimension by employing a fluid solver to simulate the full system in some interval of time and use latent evolution to evolve in other time intervals, with a predefined alternative schedule. The full system uses a uniform *fluid representation* for any snapshot of the time. In contrast, our LHPC addresses the multi-physics challenge of modeling N-body dynamical systems, which require a **heterogeneous** particle-continuum description. Our LHPC thus represents the full system with a hybrid particle-continuum representation, and addresses the long-standing challenge of modeling their coupling in a more efficient but accurate way, which cannot be done by uniform (fluid) representations. Therefore, the problems that LHPC and LED address are very different, but can certainly be complementary. Similarly, solver-in-the-loop is also based on a uniform (fluid) representation, using a neural network to correct for the fluid solver’s error, where the solver operates on a coarser spatial resolution.
> We have confirmed the effectiveness of LHPC and the importance of a hybrid (heterogeneous) representation in the experiment with the ablation of using a single representation (of either the particles or the fluid alone), with strong baselines of CNN and state-of-the-art model FNO as the base architecture, while keeping the other aspects the same. This is shown in Table 1 and 2 of the paper.
>
> We have revised Section 2 of the manuscript to make this clearer.

---

### Official Review · Reviewer_gLTB · 2022-10-25

**Confidence:** 4
**Correctness:** 4
**Technical Novelty And Significance:** 3
**Empirical Novelty And Significance:** 3
**Recommendation:** 8

**Clarity, Quality, Novelty And Reproducibility:**

The presentation of the paper is clear, but may not be easy to follow by people outside this area. The idea presented in the paper is interesting.

**Strength And Weaknesses:**

Strengths
1.	The proposed method combines classical approaches with neural networks models in addressing important scientific problems.
2.	A separate neural network is used to couple the classical solver for particle distribution and the NN model with continuum representation.

Weaknesses
1.	Why not use NN to simulate the non-thermal part? I assume a NN-based surrogate model for both thermal and non-thermal component can further accelerate the speed.
2.	It would be better to evaluate the method in more than one application.


**Summary Of The Paper:**

This paper proposes a hybrid framework for modeling N-body systems. It uses a classical solver for the non-thermal particle distribution. The neural networks are used in two components: (1) in modeling the thermal part using a continuum representation, (2) in coupling the continuum model and the particle model. The proposed method has been shown effective in modeling laser plasmas. Overall, this paper is interesting as it combines classical physical approaches with neural networks to address a challenging and important physical problem.

**Summary Of The Review:**

Overall, this paper proposes an interesting idea of combining the classical solver of physical systems and neural network models, and evaluates this algorithm on a challenging problem.
It would be more convincing to compare with (1) different design of NN models (e.g., using NN for non-thermal part as well), and (2) different NN architecture. Also, it can be more convincing to show the effectiveness in multiple physical applications.

---

> ### Author Response · Authors · 2022-11-16
> **Official Response**
>
> We thank the reviewer for recognizing the significance, novelty and clarity of the work. In the following, we address the questions/comments of the reviewer:
> > Why not use NN to simulate the non-thermal part? I assume a NN-based surrogate model for both thermal and non-thermal component can further accelerate the speed.
>
> Answer: Thank you for raising this important point.  In fact, in our original submission, our baselines of “Baseline: Bi-Gaussian” and “FNO: Bi-Gaussian” were added exactly to address this point. As we can see in Table 1, using the “Baseline: Bi-Gaussian” slightly improves the speed (from 0.115s to 0.110s). However, for the key quantities of interest, Field and M_PIC, the long-term (50-step) evolution error for the Bi-Gaussian is significantly larger. This is because the momentum distribution of the far-from-equilibrium particles is actually quite complex (as seen in Figure 6) resulting in much larger error. Given the importance of correctly capturing the details of this non-thermal distribution for many applications of interest, such as particle acceleration for material studies, medical applications, or astrophysics (e.g., cosmic-ray acceleration), this motivated retaining a first-principles particle description for this component of the distribution and learning how to couple it to the bulk distribution in a more efficient but accurate way.  These results demonstrate the advantage of our LHPC’s hybrid representation.
>
> > It would be better to evaluate the method in more than one application.
>
> Answer: We appreciate the suggestion and agree that this is a natural next step. We note that we choose to evaluate LHPC on a very challenging multi-scale, multi-physics N-body system. For example, while plasmas can exhibit the same nonlinear fluid phenomena observed in gas dynamics they are further challenged by long-range electromagnetic interactions which make the multi-scale modeling of non-equilibrium N-body phenomena considerably more complex. The laser-plasma systems that we model already push the limits of existing numerical methods, and, as such, solving it already addresses a long-standing problem. Given the performance of LHPC in such a complex problem and the general importance of particle-continuum descriptions of non-equilibrium, N-body systems (including hypersonic flow and plasma dynamics, materials science, and astrophysics) we are thus confident that this method can impact research in a wider number of disciplines. In other words, given the complexity of the problem that we target in our manuscript, we believe that our current evaluation is sufficiently general to demonstrate the effectiveness and applicability of the method. Applying it to other important applications is in our plan and will be an exciting part of the future work.

---

### Official Review · Reviewer_mprf · 2022-10-25

**Confidence:** 3
**Clarity, Quality, Novelty And Reproducibility:** It's better to give a clear definitio…
**Correctness:** 3
**Technical Novelty And Significance:** 2
**Empirical Novelty And Significance:** 3
**Recommendation:** 5

**Strength And Weaknesses:**

Strength:
+ the motivation is clear, and the proposed method seems to work.
+ The writing of the paper is generally clear. I can read the paper well though I don't have much background knowledge in physics.

Weakness:

- The novelty of the proposed method is limited in terms of machine learning techniques. This type of system is typical in machine learning.
- This work may not be able to draw much interest from the wide audience of this conference, whose main focus is on innovations in learning systems.

**Summary Of The Paper:**

This work studies a new learning system to model a non-equilibrium N-body system. In particular, the new method combines two learning components to jointly model the near-equilibrium (thermal) flow and far-from-equilibrium (non-thermal) particles. The experiments show that this hybrid model outperforms a model that only describes the system in a single mode (thermal or non-thermal).

**Summary Of The Review:**

Overall I think the main focus of this work is to solve a modeling problem in physics. But I don't have enough knowledge to judge the novelty and correctness of this contribution. On the machine learning side,  I feel that the proposed model is very typical and does not bring much innovation. I also worry that the work deviates too much from the main theme of this conference.

---

> ### Author Response · Authors · 2022-11-16
> **Official Response**
>
> We thanks the reviewer for the constructive review. Below, we address the points raised by the reviewer.
>
> > The novelty of the proposed method is limited in terms of machine learning techniques. This type of system is typical in machine learning.
>
> > This work may not be able to draw much interest from the wide audience of this conference, whose main focus is on innovations in learning systems.
>
> Answer: We thank the reviewer for their thoughtful comments. We would like to emphasize that our submission is in the area of “Machine Learning for Sciences”. This is a new area of the ICLR Conference driven by the fast-growing interest in this topic and in recognition of the very high potential for machine learning (ML) to impact scientific applications. In particular, our work addresses a long-standing problem in the modeling of non-equilibrium, N-body physical systems: how to couple particle-continuum descriptions in an efficient but accurate way. This is of wide interest as it affects a large number of disciplines, including hypersonic flow and plasma dynamics, materials science, and astrophysics. Previous efforts to address this problem have been largely based on conventional (non-ML) numerical techniques and ad-hoc models. Current solutions are incomplete and inaccurate, preventing their wider adoption and application, as discussed in our manuscript.
>
> Our work introduces a method for Learning Hybrid Particle-Continuum (LHPC) models from the data of first-principles particle simulations. The most computationally-intensive particle solver
> is used to advance the non-thermal particles, whereas a neural network solver is used to efficiently advance the thermal component using a continuum representation. Most importantly, an additional neural network learns the particle-continuum coupling: the dynamical exchange of mass, momentum, and energy between the particle and continuum representations. Training of the different neural network components is done in an integrated manner to ensure global consistency and stability of the LHPC model. While we agree that the individual components use standard networks (e.g., CNN, MLP), our innovation lies in how they are combined to address this scientific problem of wide interest. The correct coupling of the different descriptions is far from trivial as demonstrated by the large errors associated with the baseline tests using either a single NN (all-fluid) or two NNs (bi-Gaussian) as well as by the new results where we ablate the component of the coupling network, which have been included in the revised manuscript. The complex structure of the distribution function of the particles far from equilibrium (non-thermal) together with the fact that in most applications of interest this is a primary target of interest and should be very accurately described motivate the need to use a hybrid representation with a particle description for this population. The use of a NN to advance the thermal component using a continuum description provides a significant speedup given that most particles are associated with this population. The integrated training of a second NN to learn the particle-continuum coupling proved critical to accurately capture the exchange of mass, momentum, and energy between the hybrid descriptions. To the best of our knowledge the way ML techniques were combined to solve this problem is new and is our main innovation. Given the wide importance of particle-continuum descriptions in the physical sciences and the interest of ICLR on ML for Sciences we strongly believe that this work will impact research in different scientific domains and will be of interest to the ICLR community.
>
> > It's better to give a clear definition or description of the loss function.
>
> Answer: We thank the reviewer for the suggestion. In the Appendix E of the original submission, we had briefly stated the training procedure and loss function used. In the updated draft, we have expanded Appendix E to give a more detailed description of training and loss function. We have also updated the main text to directly point to Appendix E in Section 4.2 when discussing the loss function. Please refer to the updated Appendix E for the detailed equation of the loss function.

---

> > ### Comment · Reviewer_mprf · 2022-11-19
> > **Thank you for your response**
> >
> > Thank you for your detailed response. I agree that the work is innovative in the problem of modeling non-equilibrium, N-body physical systems. I will calibrate my evaluation of the work after I see more discussions.

---

> > > ### Author Response · Authors · 2022-12-07
> > > **Response**
> > >
> > > Dear Reviewer mprf, we thank you for recognizing the novelty of our work. As you know, particle-continuum coupling is a long-standing problem in modeling dynamical systems, and we believe it will be of interest to the ICLR community due to the growing focus on AI for Science. We would like to take this opportunity to bring to your attention two recent papers (from NeurIPS this year) that we hope can further help calibrate your evaluation:
> > >
> > > * MAgNet: Mesh Agnostic Neural PDE Solver (https://arxiv.org/abs/2210.05495)
> > > * M2N: Mesh Movement Networks for PDE Solvers (https://arxiv.org/abs/2204.11188)
> > >
> > > Many recent papers have focused on adaptive meshing or interpolation on meshes to more efficiently solve PDEs, but such methods are based on a single continuum representation which can fundamentally breakdown at small scales, particularly for far-from-equilibrium systems. Our method can be considered a complementary innovation to such methods, and absolutely necessary, in multi-scale and multi-physics systems where evolving individual particles in far-from-equilibrium regions with a high-fidelity solver is required. These systems underly important applications, from compact ion accelerators for cancer radiotherapy, hypersonic aircrafts, soft materials, modeling of high-energy astrophysical events, and much more. Previously, the efficient modeling of these systems has been stymied across many domains and applications due to the absence of an adequate solution for particle-continuum coupling. Since our innovation revolves around a hybrid representation with a deep learning solution to couple the two representations, we believe ICLR, with its emphasis on representation learning and its growing focus on AI for the natural sciences, is the ideal venue for our work.

---

### Author Response · Authors · 2022-11-16
**General Response**

We thank the reviewers for their constructive comments. We are glad that the reviewers generally recognize our method’s significance as well as the clarity of the paper and the soundness of the work. Based on reviewers’ valuable feedback, we have performed additional experiments and updated the manuscript to address the reviewers’ concerns. The main additional experiments and improvements are as follows:

1. We discuss in the response to the reviewer mprf the significance and novelty of our method. In summary, we believe that this work addresses a long-standing problem in the modeling of non-equilibrium N-body, physical systems: how to couple particle-continuum descriptions in an efficient but accurate way. The way machine learning is used to address this problem is new and this innovation will impact research in different scientific domains, thus being of interest to the ICLR community and the wide effort in Machine Learning for Sciences;

2. We have significantly expanded Appendix E and detailed the training procedure and loss function used for training our LHPC;

3. According to the suggestion of reviewer qT1Y, we added the ablation of “LHPC (no-coupling)”, which ablates the component of the coupling network. Together with other ablations in the original submission, they show that all the three components are essential for improving the long-term accuracy and speed of the simulation and further strengthen the importance of our contributions;

4. We have updated Section 2 “Related works” where we elaborated further on the importance of a hybrid continuum-particle representation and on how previous ML-based works are primarily based on uniform representations which can be quite powerful for other systems, but are not appropriate for far-from-equilibrium N-body systems.

---

### Decision · Program_Chairs · 2023-01-20

**Decision:**

Reject

**Justification For Why Not Higher Score:**

The novelty on the technical side seems limited, and the work may not be able to draw much interest from the wide audience of ICLR.

**Justification For Why Not Lower Score:**

Experimental results show that this hybrid model outperforms a model that only describes the system in a single mode (thermal or non-thermal). The motivation is clear and the paper is nicely written.

**Metareview: Summary, Strengths And Weaknesses:**

This paper proposes a hybrid framework to model a non-equilibrium N-body system. The new method combines two learning classical components to jointly model the near-equilibrium (thermal) flow and far-from-equilibrium (non-thermal) particles. Experimental results show that this hybrid model outperforms a model that only describes the system in a single mode (thermal or non-thermal). The motivation is clear and the paper is nicely written. However, the novelty on the technical side seems limited, and the work may not be able to draw much interest from the wide audience of ICLR.